# Mechanical and Thermal Properties and Moisture Sorption of Puffed Cereals Made from Brown Rice, Barley, Adlay, and Amaranth

**DOI:** 10.3390/foods14020189

**Published:** 2025-01-09

**Authors:** Atsuko Takahashi, Keiko Fujii

**Affiliations:** Department of Food Science and Nutrition, Faculty of Human Sciences and Design, Japan Women’s University, 2-8-1 Mejirodai, Bunkyo-ku, Tokyo 112-8681, Japan; jwu00010722@fc.jwu.ac.jp

**Keywords:** puff, cereal, moisture sorption, glass transition, rupture properties

## Abstract

The moisture sorption, rheological, and glass transition properties of puffed cereals, such as brown rice, barley, adlay, and amaranth, were assessed. The puffed cereals were stored in desiccators until their moisture content reached equilibrium. Moisture sorption isotherms were measured, and monomolecular adsorption moisture content was calculated through Brunauer−Emmett−Teller (BET) analysis. The glass transition temperature (Tg) was determined, and the internal structure was observed using a scanning electron microscope. The rupture force and apparent elastic modulus of puffed cereals decreased with increasing relative humidity (RH). The puffed cereals exhibited ductile fracture ,when the moisture content was >8%. The Tg of puffed cereals with 8% moisture content was approximately 40 °C. It was inferred that puffed cereals demonstrated a crispy texture in the glassy state when stored at <40 °C, but transitioned to a rubbery state at >40 °C, resulting in the loss of crispy texture.

## 1. Introduction

Because food is a multi-component system, microbial spoilage and deterioration owing to reactions can occur during transportation and storage. Therefore, various attempts have been made to protect food from deterioration. In food engineering, the development of methods to reduce temperature and moisture and improve shelf life is vital [1].

The curve plotting the equilibrium moisture content against various relative humidities at a certain temperature is called a moisture sorption isotherm. It is useful for analyzing storage and processing properties by predicting the moisture adsorption and drying of foods in a certain environment. Additionally, they can be used to classify food groups based on similarities in their behavior. Numerous foods are known to exhibit an inverted S–shape moisture sorption isotherm.

By determining the monomolecular adsorption moisture content from the region of strong moisture sorption that occurs at the rising edge of the moisture sorption isotherm curve, it was observed that microbial growth was inhibited, enzymatic and browning reactions were minimized, and numerous degradation reactions of food products were inhibited [1,2,3,4,5,6,7]. Previous studies using moisture sorption isotherms include those that examined the moisture sorption characteristics of biscuits [8] and wafers [9] and those that examined the relationship between moisture sorption and texture characteristics of various baked confections [10,11,12,13,14,15,16,17].

Additionally, numerous dried foods are in a glassy state, resulting in research on quality control that considers glass transition [18,19]. The focus on glass transition stems from its ability to explain the phenomena and rupture properties associated with food deterioration and shelf life.

Previous studies have revealed the moisture sorption and plasticity in moisture-sorbed snacks, corn cakes [20], and wafers [9] by exploring their glass transition points. However, despite 30 years of research on glass transition in food, few studies have been conducted on puffed cereals.

Therefore, this study aimed to clarify the moisture sorption and rheological properties of puffed cereals by examining the effects of moisture absorption on their rupture properties. This includes assessing alterations in the breaking behavior, monomolecular adsorption moisture content, and glass transition point of the sorbed puffed cereals. These aspects are characteristic of swollen foods.

## 2. Materials and Methods

### 2.1. Experimental Materials

Four types of cereals with characteristic nutritional functions and varying grain sizes were selected as samples for various puffed cereals: brown rice (domestic), barley (American), adlay (Iwate), and amaranth (Iwate). The long and short diameters of the four types of cereals (raw) are as follows. Brown rice: 4.88 mm, 2.85 mm; barley: 4.06 mm, 2.98 mm; adlay: 5.15 mm, 4.72 mm; amaranth: 1.16 mm, 1.16 mm.

Of the four cereals, brown rice, adlay, and amaranth were purchased from the Japan Millet Association, and barley was provided by Mitake Co. (Saitama, Japan). The cereals were swelled at the Sekizawa Shouten Ltd. (Saitama, Japan) swelling and processing plant in the presence of the authors.

The nutritional components of the four cereals in 100 g are shown in Table 1 [21].

### 2.2. Preparation of Puffed Cereals

The swelling of various cereals was investigated using a cereal puffing machine (Koyo Machine Manufacturing Co., Ltd. Hiroshima, Japan). A pressure kiln was preheated to 230 °C, and 2 kg of grain was placed inside the kiln. The pressure was increased to 0.1, 0.7, 0.9, and 1.1 MPa and was reduced rapidly to induce swelling. Briefly, raw samples (grains) were placed in a pressure kiln and heated at high pressure while rotating, and when a certain pressure was reached, the valve of the pressure kiln was struck with a hammer, the lid was opened, and the internal pressure was released at once. The samples swelled while retaining the shape of the grain. The heating temperature was 240 ± 20 °C, and the heating time was approximately 1 min for the 0.1 MPa treatment and 5–7 min for the 0.7–1.1 MPa treatment. The samples were stored in eight humidity-controlled containers with relative humidity (RH) ranging from 6 to 94% for 5–12 days until their moisture sorption reached equilibrium. The humidity control temperatures were 15 °C, 25 °C, and 35 °C. For humidity control, eight different saturated aqueous solutions of salts were used: lithium bromide for 6.0–6.9% RH; lithium chloride for 11.3% RH; magnesium chloride for 32.1–33.3% RH; potassium carbonate for 43.2% RH; sodium bromide for 54.6–60.7% RH; sodium chloride for 74.9–75.6% RH; potassium chloride for 83.0–85.9% RH; and potassium nitrate for 90.8–95.4% RH.

### 2.3. Measurement of Moisture Content and Equilibrium Moisture Content

The moisture content of the moisture-sorbed puffed cereals (1.5–3.0 g) was measured. A moisture analyzer (MB45; OHAUS Co., Ltd. Tokyo, Japan) was used to measure the moisture content. The samples were heated at 105 °C until the rate of weight change reached 1 mg/120 s.

The equilibrium moisture content was calculated using the following equation:Equilibrium moisture content (% dry basis [D.B.]) = {Weight at equilibrium moisture sorption × Moisture content at moisture sorption equilibrium/Weight at equilibrium moisture sorption × (1 − moisture content before moisture sorption)}(1)

### 2.4. Moisture Sorption Isotherm and Measurement of Monomolecular Adsorption Moisture Content

In many foods, the moisture sorption isotherm exhibits an inverted S-shape. A typical moisture sorption isotherm can be divided into three regions, as shown in Figure 1. In the region with the lowest water activity, moisture molecules are adsorbed on the food surface, forming a monomolecular layer. In this zone, moisture molecules are restricted, minimizing physical and chemical alterations and significantly reducing the deterioration rate of the food. Because the BET formula is often used to determine the monomolecular adsorption moisture content, it was calculated by fitting the values of each moisture sorption isotherm in the 5–35% RH range to the BET formula [4].

Monomolecular adsorption moisture content was calculated based on the parameter analysis of the BET equation, as follows:BET formula (Brunauer−Emmett−Teller) A/(V(1 − A)) = (C − 1)/CVm·A + 1/(CVm)(2)

A: Water activity of food sampleV: Moisture content of food sample (g/100 g dry matter)Vm: Monomolecular adsorption moisture content (g/100 g dry matter)C: Constant determined by the properties of the food sample and temperatureA/V(1 − A) = A·(C − 1)/C·Vm + 1/C·VmIf A/V(1 − A) = y, A = x, (C − 1)/C·Vm = a, and 1/C·Vm = bThis can be expressed by the linear equation y = ax + b.a = (C − 1)/C·Vm... Slope of the straight lineb = 1/(C − 1)...y–interceptC = (b + a)/bVm = 1/(b + a)∴ Vm: monomolecular adsorption (g/100 g dry matter) was calculated∗ The BET formula reportedly fits well in the 0.05–0.35 water activity range.

### 2.5. Rupture Properties of Puffed Cereals Following Moisture Sorption

The rupture properties of various moisture-sorbed puffed cereals were measured with a rheometer (RE-3305; Yamaden Co., Tokyo, Japan) using the single-grain method [22,23]. An acrylic resin plunger with a diameter of 40 mm was used. The compression ratio was 99%, and the compression speeds were 0.05, 0.1, 1, and 10 mm/s at the four levels. The rupture force and apparent elastic modulus were calculated from the resulting force-deformation curves. For the amaranth specimens, automatic creep meter analysis device software Rupture Strength Analysis (HS Windows Ver. 2.5; BAS-3305H; Yamaden Co., Ltd.) was used at a compression rate of 10 mm/s. The measurement temperature was 25 °C and the measurements were taken three times.

### 2.6. Measurement of Glass Transition Temperature (Tg)

The Tg of the puffed cereals was measured using a differential scanning calorimeter (Diamond DSC; Perkin Elmer Japan Co., Kanagawa, Japan). The samples were ground in a mortar, and 7–42 mg in weight was placed in a large stainless steel pan. The measurement temperature ranged from −50 °C to 100 °C, and the scan speed was 10 °C/min. Two scans were obtained for each sample.

### 2.7. Observation of Tissue Structure

Various puffed cereals were sliced into approximately 1 mm–thick slices, and platinum–palladium deposition was performed. The tissue structure was observed using a scanning electron microscope (S-800; Hitachi, Ltd., Tokyo, Japan) [24] at 40×, 100×, and 500× magnifications.

### 2.8. Statistical Processing

SPSS (Ver. 25.0; IBM Japan, Ltd., Tokyo, Japan) was used for statistical processing and analysis. Tukey’s multiple comparisons were performed after one-way analysis of variance (ANOVA) to test variations in puffed cereals between samples. In all cases, a risk rate of <5% was set as the significance level.

## 3. Results and Discussion

### 3.1. Moisture Sorption Characteristics

Various types of puffed cereals were humidified, and their equilibrium moisture content was determined. Subsequently, the moisture sorption isotherms for each cereal puff were plotted, with RH on the horizontal axis and equilibrium moisture content on the vertical axis (Figure 2). In this study, the majority of the moisture sorption isotherms exhibited an inverted S-shape, except for a few isotherms.

The moisture sorption isotherms of various puffed cereals at 0.1 MPa and amaranth puffs at 0.7 MPa indicated that all samples exhibited moisture content < 3% D.B. at 15% RH. None of the samples were swollen. However, the other swollen samples exhibited a higher moisture content of approximately 6% D.B. at 15% RH. This indicates that moisture sorption increases significantly in the region where the isotherm curve rises. Thus, it was inferred that sorption was stronger in puffed cereals with advanced swelling.

Figure 3 shows the moisture sorption isotherms at different temperatures for various puffed cereals swelled at 0.9 MPa. Up to 80% RH, there was no difference in moisture content between the puffed cereal samples. However, at 15 °C and approximately 90% RH, the moisture content increased, specifically for barley and adlay, reaching approximately 32% D.B. At 25 °C and 35 °C, the moisture contents of brown rice, barley, and adlay at approximately 90% RH were similar. Conversely, for amaranth, the moisture content increased as the humidity conditioning temperature decreased (35 °C < 25 °C < 15 °C). This aligns with the general inference that the moisture content decreases at higher temperatures [1], likely owing to the temperature dependence of the adsorption potential function [2] and alterations in intermolecular forces between the solid and the adsorbent.

### 3.2. Monomolecular Adsorption Moisture Content

The moisture sorption isotherms of various puffed cereals humidified at 15 °C, 25 °C, and 35 °C exhibited an almost inverse S-shape. 

Table 2 presents the results of the amount of monomolecular adsorption for all the samples at 15 °C, 25 °C, and 35 °C. As shown, the monomolecular adsorption moisture content of puffed cereals was approximately 10–18 g/100 g dry solid, which is higher than that of the 3.7–4.7 g/100 g dry solid observed in similar dried foods, such as cookies, biscuits, and snacks [25]. Foods with relatively similar values were 11 g/100 g dry solid for stick agar and 14 g/100 g dry solid [25] for kanpyo. Wada et al. [25] studied the effect of absorbed moisture on the textural properties of low-moisture foods and observed that samples with high monomolecular adsorption had high starch content and low oil and sugar content. In the present study, the results in Table 2 show that amaranth has the lowest value at all temperatures. Looking at the nutritional compositions of the four types of puffed cereals in Table 1, amaranth had the lowest value for carbohydrates and the highest value for fat among the four species. This could be the reason for the lower monomolecular adsorption of amaranth.

### 3.3. Rupture Properties of Puffs After Moisture Sorption (Rupture Force)

Figure 4 shows the relationship between the equilibrium moisture content and rupture force of various puffed cereals swollen at 0.9 MPa. In all samples, the rupture force increased to approximately 8% of the equilibrium moisture content. However, at higher equilibrium moisture contents, the rupture force decreased as the equilibrium moisture content increased. This aligns with numerous studies [10,20,25,26,27,28,29,30,31,32,33] that have demonstrated that stress in moisture-sorbed foods increases or decreases with increasing equilibrium moisture content. This is attributed to the fact that starch binds and becomes more viscous with moisture absorption, which increases its rupture properties. However, in the higher humidity range, the adsorbed moisture molecules cause flow deformation, which reduces the rupture properties [32]. In the present study, the reduction in the rupture force at high equilibrium moisture content in the 15 °C humidity-controlled samples was attributed to the moisture sorption isotherm (Figure 3). This demonstrates that the equilibrium moisture content in all four types of puffed cereals was significantly higher at 15 °C humidity, resulting in water acting as a plasticizer and causing flow deformation.

The significantly increased rupture force of puffed brown rice at 15% equilibrium moisture content was attributed to the shrinkage and tightening of the samples in the rubbery state with water.

The relationship between the amount of monomolecular adsorption and the rupture force was also examined. The amount of monomolecular adsorption is obtained in the 0.05–0.35 water activity range, which is approximately 4–8% at equilibrium moisture content. In Figure 4, the values of rupture force at 4–8% equilibrium moisture content are approximately 8–20 N for brown rice and adlay, 18–40 N for barley, and 2–4 N for amaranth, with amaranth having the lowest value. The values of monomolecular adsorption at 0.9 MPa in Table 2 show that amaranth has the lowest values at all temperatures (15 °C, 25 °C, and 35 °C).

Figure 5 shows the force-deformation curves of brown rice at 25 °C. The ductile fracture was observed from RH 93.6, 57.6, 57.6, and 32.8% RH at 0.1, 0.7, 0.9, and 1.1 MPa swelling, respectively. The dotted line represents the borderline between brittle and ductile fractures, and the filled area represents the ductile fracture region. The higher the swelling pressure, the lower the RH in the boundary region between the brittle and ductile fractures. Similar trends were observed for the three types of puffed cereals, excluding brown rice; therefore, the results for brown rice were considered representative.

Figure 6 shows the moisture sorption isotherms of the four types of puffed cereals. The transition from brittle to ductile fracture is indicated by the moisture sorption isotherm, with the circled region indicating the boundary. All the samples with swelling pressures between 0.7 and 1.1 MPa exhibit ductile fracture when the moisture content was approximately >8%. This demonstrates that the crispiness of puffed cereals can be effectively assessed by identifying the brittle to ductile transition in the fracture curve, which occurs at a moisture content of 8%. In a previous study [34], at water contents higher than 8.5% for CCF (commercial corn flakes), the compression curves showed fewer fluctuations, reflecting a decrease in brittle fracture. Similarly, in the present study, the fracture curve changed from brittle to ductile fracture at >8% moisture content.

The products produced by baking or extruder exhibit a water activity value of 0.5, or an equilibrium moisture content of approximately 8–10%. Additionally, the critical moisture content for crispiness was in the water activity value range of 0.45–0.55 [35]. In the present study, samples other than amaranth with swelling pressures of 0.7–1.1 MPa exhibited brittle fracture up to 43.2% RH and ductile fracture from 57.6%RH, similar to previous studies. Amaranth was the only exception, with 0.7 MPa showing a brittle fracture up to 57.6%RH and a ductile fracture from 75.3%RH. According to a previous study [36], the 0.7 MPa amaranth sample had a low swelling ratio of 1.13, which was close to that of the raw sample. In other words, the 0.7 MPa amaranth sample with insufficient swelling was considered to be in a glassy state, while the other samples with advanced swelling absorbed water and became rubbery, and therefore, the brittle fracture zone of amaranth was under 75.3%RH.

This indicates that the crispiness of puffed cereals can be assessed effectively by the brittle and ductile fractures in the force-deformation curve, with brittle fractures transitioning to ductile fractures beyond 8% moisture content.

### 3.4. Microstructure of Puffed Cereals

Figure 7 shows cross-sectional photographs of the internal microstructure of various puffed cereals at 90.8–95.4% RH humidification. Numerous large pores were observed in brown rice, barley, and adlay at 25 °C and 35 °C humidification, whereas smaller pores were observed at 15 °C humidification. In amaranth, the pores became smaller with decreasing temperature (35 °C > 25 °C > 15 °C).

Figure 3 shows that the equilibrium moisture content at 25 °C and 35 °C did not increase with an increase in relative humidity, but the moisture content at 15 °C increased significantly. This was thought to be due to the cell walls of the puffed cereals becoming thicker and the pores becoming smaller as a result of water absorption. Conversely, the equilibrium moisture content of the amaranth sample at approximately 90% RH was 35 °C < 25 °C < 15 °C, and the equilibrium moisture content increased as the temperature decreased, which is consistent with the pore size in the electron micrograph shown in Figure 7.

Figure 8 shows the relationship between the internal microstructure and force-deformation curve during the compression of four types of puffed cereals swollen at 0.9 MPa and humidified at 25 °C. When the puffs were moistened at 6.7% RH, the pores were relatively large, and the continuous phase was significantly thin. The force-deformation curve exhibited a clear break point, indicating that the puffs were brittle with a low rupture force. The internal microstructure at 43.2 and 75.3% RH demonstrated that the pore size did not change much. However, the continuous phase constituting the pores gradually became thicker as the RH increased.

The force-deformation curve at 43.2% RH demonstrated that the reduction in force became smaller after fracture, and at humidity control > 75.3% RH, the force-deformation curve exhibited a ductile fracture. At 93.6% RH, the pores were smaller, and the continuous phase was considerably thicker. The force-deformation curve exhibited a significant increase in force with slight deformation, indicating that the moist specimen shrunk during compression and did not fracture.

The pore size of barley was different from that of brown rice, adlay, and amaranth, which had fibrous pores. Because of this shape, the cell walls were thicker, and the angle of the rising curve at the beginning of compaction was greater.

### 3.5. Measurement of Tg

Determining Tg is critical for determining the shelf life of food. At temperatures below Tg, Brownian micromotion and molecular migration are reduced, thereby stabilizing food quality and enhancing shelf life. Conversely, above Tg, molecules exhibit enhanced Brownian micromotion and greater molecular migration, leading to deterioration of the food product. Therefore, knowing the Tg is critical for identifying the temperature at which the food product can be stored to maintain quality and extend its shelf life.

The Tg values of various puffed cereals conditioned at 25 °Care illustrated in Figure 9. The Tg ranged from −16.6 °C to 80.2 °C. Tg decreased as the equilibrium moisture content increased, which is consistent with the trends for corn cakes [20], wafers [9], corn flakes [34], and extruded cereal-based products [37].

In the previous section, it was demonstrated that the boundary region where the rupture properties transition from brittle to ductile fracture occurs at a moisture content of 8%. Additionally, it was identified that the Tg of these puffed cereals with 8% moisture content was approximately 40 °C. These puffed cereals exhibited a crispy texture in a glassy state when stored at 25 °C. However, it was inferred that they shifted to a rubbery state at temperatures above 40 °C, and the crispy texture was lost.

In a previous study [37], Young’s modulus results showed that water acts as an anti-plasticizer at low aw, while exhibiting a plasticizing effect at high aw, and a stability map can explain the brittle-ductile transition that occurred when it was below Tg. In the present study, as in the previous studies [34,37], the threshold between brittle and ductile fracture was almost at Tg.

Figure 10 shows the effect of moisture content on the apparent elastic modulus of various puffed cereals. In all samples, the apparent elastic modulus decreased as the moisture content increased.

Based on a previous study that assessed the effect of moisture adsorption on the rupture properties of commercial confectioneries [32], the apparent elastic modulus decreased with increasing humidity, and the transition was relatively small up to an RH of approximately 43%; however, it decreased significantly at higher RH. Similar results were obtained in this study. This is attributed to the glass transition of various puffed cereals as a result of moisture content, which causes their rupture properties to transition to softness. The moisture content at which the apparent elastic modulus decreased rapidly was approximately 8–10% in all cases. In this study, it was assumed that puffed cereals transitioned from glass to rubber.

There are certain exceptions. For brown rice, adlay, and amaranth (excluding barley), the apparent elastic modulus was significantly higher under the highest moisture content conditions at 25 °C and 35 °C. The significantly higher apparent elastic modulus was attributed to the shrinkage and tightness of the samples in the rubbery state containing moisture.

## 4. Conclusions

Four types of puffed cereals were stored under different temperatures (15 °C, 25 °C, and 35 °C) and humidity (6–94%) conditions. The samples stored at 25 °C and 35 °C were not affected by relative humidity (RH), whereas those stored at 15 °C had higher moisture content at higher humidity. Electron micrographs of the internal structure of the samples stored at 15 °C showed that the cell walls became thicker, and the pore size of the puffs became smaller due to moisture absorption. The rupture force at this time showed a low value, indicating that the puffs were in a rubbery state following moisture absorption. The rupture force peaked at 8% RH and then decreased with an increase in relative humidity.

The physical properties changed from brittle to ductile fracture around a moisture content of 8%, and the transition point between glass and rubber was considered to be 8%.

The Tg of the puffed cereals at 8% moisture content was approximately 40 °C. It was inferred that the puffed cereals were glassy and crispy when stored at <40 °C, but changed to rubbery and lost their crispy texture when stored at >40 °C.

The results indicated that a crispy texture can be maintained by keeping the moisture content at <8% and the storage temperature at <40 °C.

## Figures and Tables

**Figure 1 foods-14-00189-f001:**
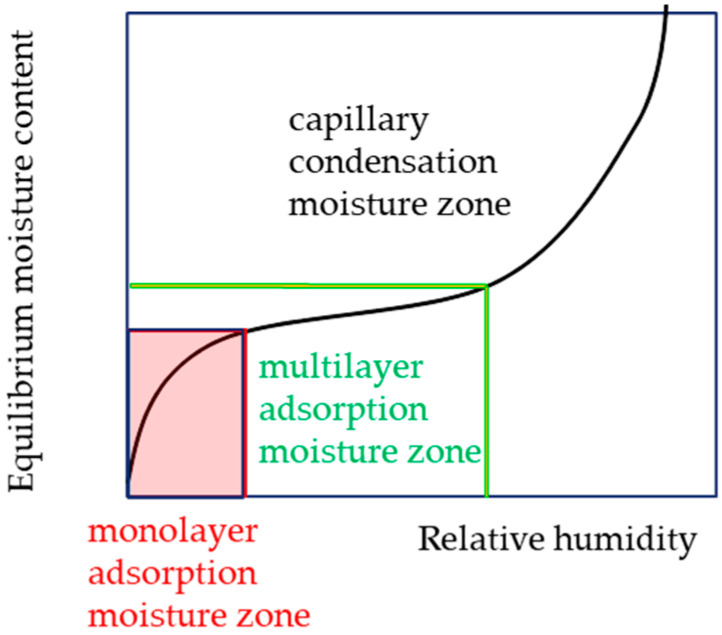
Moisture sorption isotherm divided into three regions.

**Figure 2 foods-14-00189-f002:**
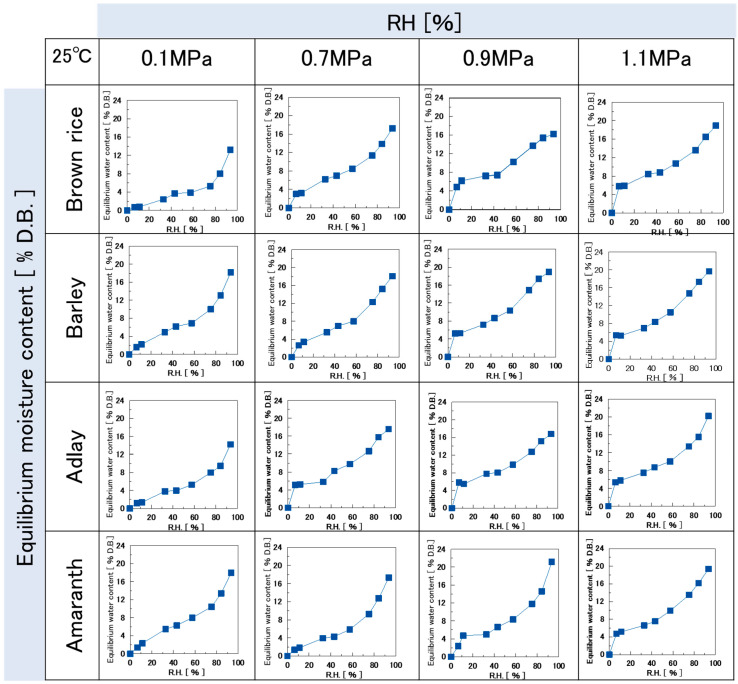
Moisture sorption isotherms of each puffed cereal at 25 °C.

**Figure 3 foods-14-00189-f003:**
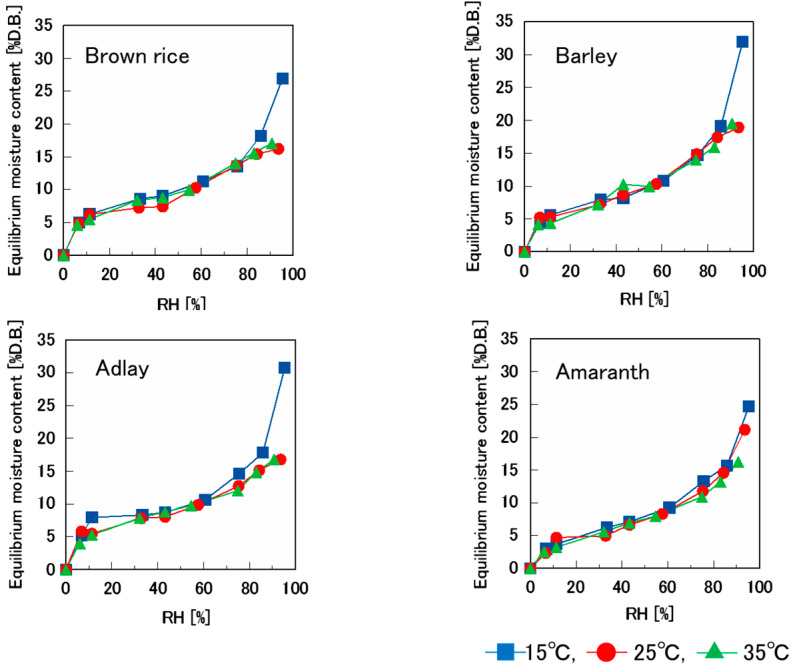
Moisture sorption isotherm of each puffed cereal of 0.9 MPa at 15 °C, 25 °C, and 35 °C.

**Figure 4 foods-14-00189-f004:**
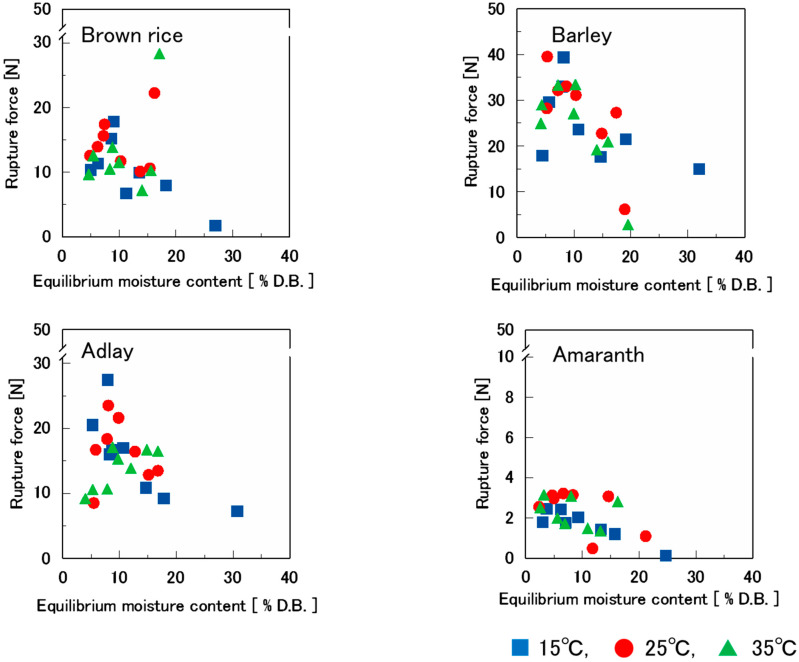
Effect of equilibrium moisture content on the rupture force of four types of puffed cereals at 15 °C, 25 °C, and 35 °C.

**Figure 5 foods-14-00189-f005:**
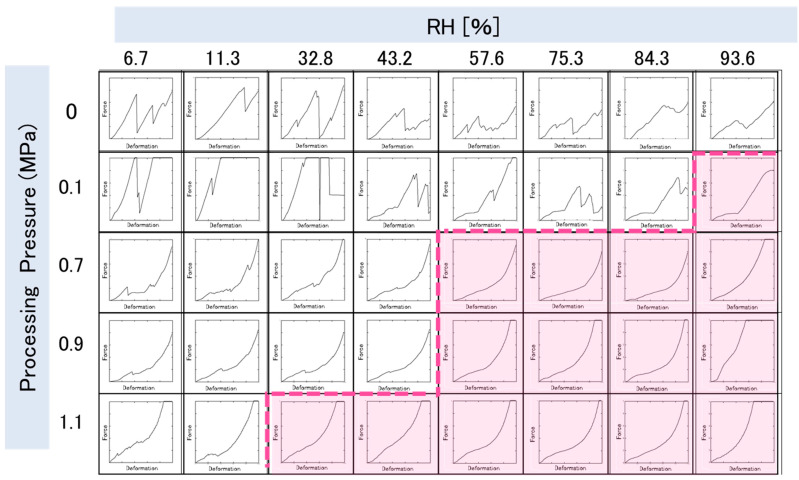
Force-deformation curves of puffed brown rice at various relative humidity (RH).

**Figure 6 foods-14-00189-f006:**
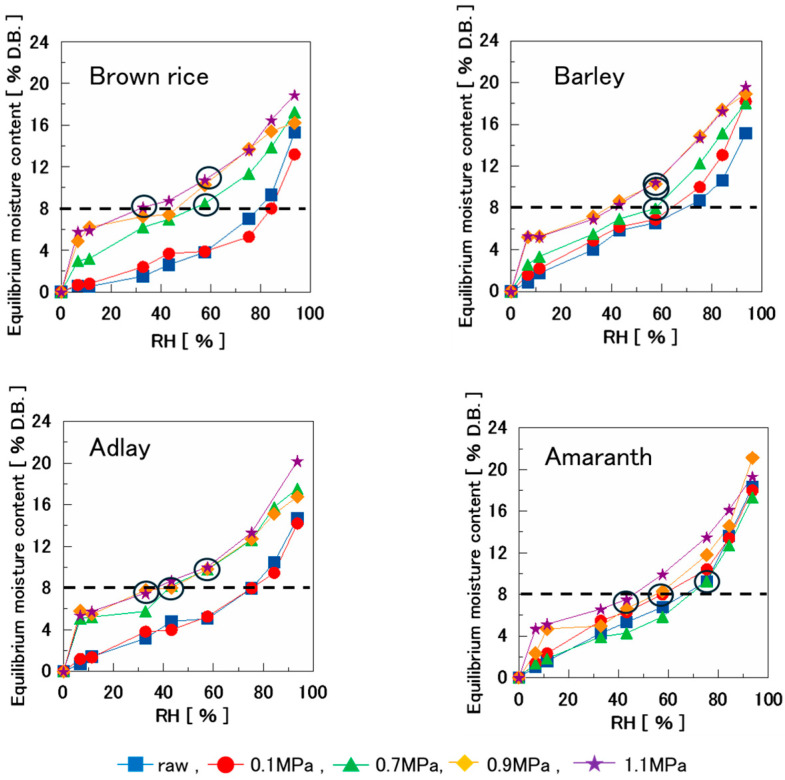
Effect of relative humidity on the equilibrium moisture content of four types of puffed cereals at 25 °C.

**Figure 7 foods-14-00189-f007:**
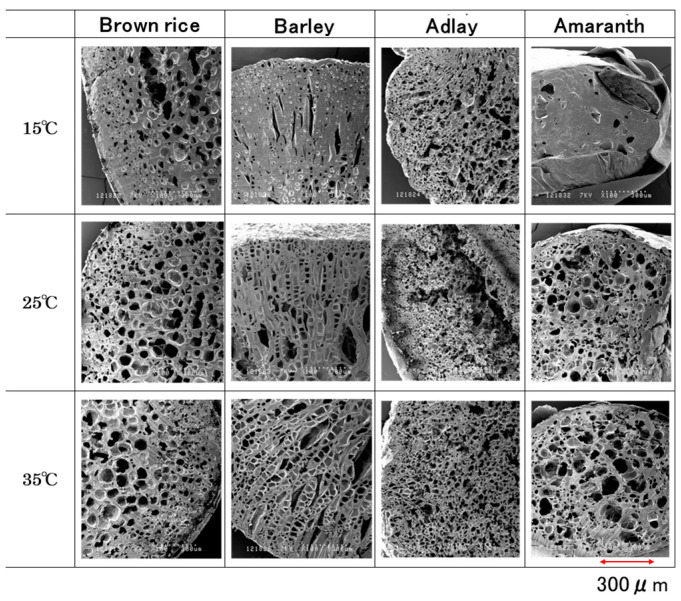
Scanning electron microscope images of the microstructure of four kinds of puffed cereals of 0.9 MPa at high humidity (100×).

**Figure 8 foods-14-00189-f008:**
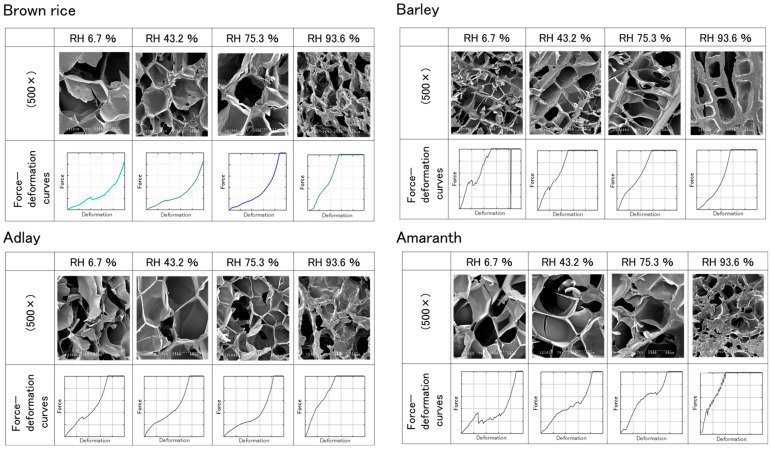
Scanning electron microscope images of microstructure and force-deformation curves of four types of puffed cereals (0.9 MPa/25 °C).

**Figure 9 foods-14-00189-f009:**
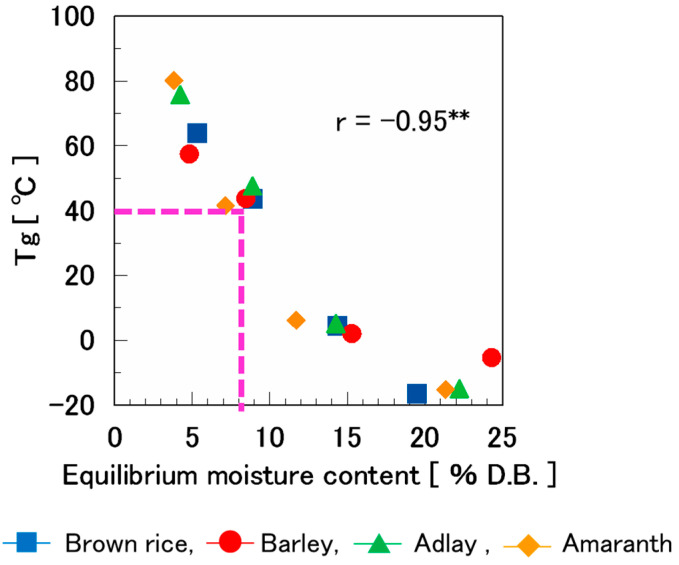
Tg in the second scan among the four types of puffed cereals (25 °C). r = −0.95 **, ** correlation is significant at *p* < 0.01.

**Figure 10 foods-14-00189-f010:**
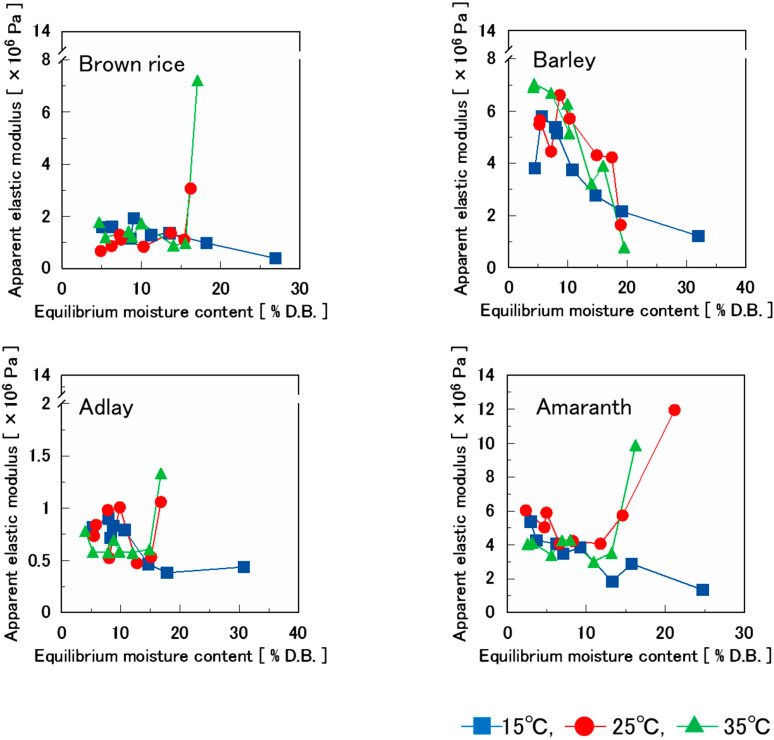
Effect of equilibrium moisture content on the apparent elastic modulus of four kinds of puffed cereals.

**Table 1 foods-14-00189-t001:** Nutritional composition of the four types of cereals.

	Energy (kcal)	Protein (g)	Fat (g)	Carbo- Hydrate (g)	Potassium (mg)	Calcium (mg)	Iron (mg)	Dietary- Fiber (g)
Brown rice	356	6.5	3.3	74.3	160	13	1	3.1
Barley	343	7	2.1	76.2	170	17	1.2	8.7
Adlay	360	13.3	1.3	72.2	85	6	0.4	0.6
Amaranth	358	12.7	6	64.9	600	160	9.4	7.4

**Table 2 foods-14-00189-t002:** Monomolecular adsorption moisture content of each puffed cereal [g water/100 g dry solid].

		0.7 MPa	0.9 MPa	1.1 MPa
15 °C	Brown rice	14.6	17.3	16.6
	Barley	14.7	16.3	17.2
	Adley	15.6	15.3	15.8
	Amaranth	14.4	14.6	13.6
25 °C	Brown rice	15.1	13.2	16.0
	Barley	12.7	13.2	12.5
	Adley	9.72	14.4	13.7
	Amaranth	13.0	10.2	17.8
35 °C	Brown rice	12.3	16.8	13.6
	Barley	16.5	14.5	13.9
	Adley	16.1	16.2	13.2
	Amaranth	10.6	12.7	13.8

## Data Availability

The data presented in this study are available on request from the corresponding author.

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
