# Peer review of "Mechanical and Thermal Properties and Moisture Sorption of Puffed Cereals Made from Brown Rice, Barley, Adlay, and Amaranth"

_foods, 2025, doi:10.3390/foods14020189_

Round 1
Reviewer 1 Report
Comments and Suggestions for Authors
The article “Moisture sorption and rheological properties of puffed cereals” has been reviewed and the following comments have been made to improve it.
Review line 26, is drying used to reduce temperature?
this work can help reinforce your introduction and dicussion: https://doi.org/10.1111/jfpp.15016
Did you process the puffed cereals? Did you buy them? Or under what conditions were they processed and purchased..
Section 2.1 Put the nutritional components in a table for a better appreciation
Section 2.2 This part needs to be detailed more, the description of the device and the procedure is very brief. Some diagrams may be useful
Line 83. The moisture analyzer model seems to be from the OHAUS brand
Line 88 ..this description is not necessary
Line 90. Review unimolecular or “monomolecular”
Section 3.2, compare with other related research and try to link with section 3.3
Figure 6 needs to be related to isotherms and pore size..as well as moisture adsorption mechanisms
Is it necessary to discuss the importance of determining Tg, why it is important, what practical uses does this information have?
Better structure the conclusions is needed
Reviewer 2 Report
Comments and Suggestions for Authors
Manuscript Title - Moisture sorption and rheological properties of puffed cereals
Manuscript Number - foods-3250399
This work explored the moisture sorption and rheological properties four kinds of puffed cereals (brown rice, barley, adlay, and amaranth) through BET, Tg, and SEM during storage at different RH and temperatures. The authors claimed following: the rupture force and apparent elastic modulus of puffed cereals decreased with increasing RH. When moisture content was above 8%, the puffed cereals exhibited ductile fracture. The Tg of puffed cereals with 8% moisture content was approximately 40 oC. It was inferred that puffed cereals demonstrated a crispy texture in the glassy state when stored at 25 oC, but transitioned to a rubbery state at temperatures above 40 oC, resulting in the loss of crispy texture. After reding the manuscript thoroughly, it gave me an impression of too superfluous and immature presentation of the literatures, results and discussion. The manuscript needs thorough revision and re-writing from a professional manuscript writer to improve its content and quality. Following points are noted during my revision process:
Title is not clear enough to indicate what this study is about. Please make it more elaborative.
Keywords should be confined to 5-6 words, representing the whole manuscript. Please follow journal guidelines for the same.
Line 57: Varying grain sizes – where is this grain size information?
Equation 1 and 2: Please insert equation function to write an equation. The present form is very complicated to understand.
Line 86: “Final: moisture sorption equilibrium”? Please mention clearly.
Fig. 1 is too small to read / view.
Line 157-160: The results presented here are very redundant and no consistent to the different treatment groups – “Table 1 presents the results, for the samples at 15, 25, and 35 °C, the values for the amount of unimolecular adsorption ranged from 13.6– 17.3, 9.72–17.8, 10.6–16.8, respectively. In this study, the unimolecular adsorption moisture content of puffed cereals was approximately 10–18 g/100 g dry solid,…”
Section 3.3: The description in this section is all about “moisture content” but the figure related to this section (Fig. 3) is about “equilibrium moisture content”. These two terminologies are completely different, I think?
Fig. 4, 5 and 7: Why the authors presented only the results of brown rice in these figures? And why they chose to skip the results of other cereal products?
English language and grammar need thorough revision from a professional English revision service provider. It is difficult to understand the messages the authors want to deliver. The authors may choose MDPI English Editing service for the same.
Comments on the Quality of English LanguageExtensive English edition is recommended.
Round 2
Reviewer 2 Report
Comments and Suggestions for Authors
Manuscript Title - Moisture sorption and rheological properties of puffed cereals
Manuscript Number - foods-3250399
The authors have addressed all the issues raised during previous round of revision. They even made changes the title to make it more elaborative to cover the study area. I have no further issue for this manuscript.
